# Deep Learning as a Mixed Convex-Combinatorial Optimization Problem

**Abram L. Friesen and Pedro Domingos**
Paul G. Allen School of Computer Science and Engineering
University of Washington
Seattle, WA 98195, USA
`{afriesen,pedrod}@cs.washington.edu`

## Abstract

As neural networks grow deeper and wider, learning networks with hard-threshold activations is becoming increasingly important, both for network quantization, which can drastically reduce time and energy requirements, and for creating large integrated systems of deep networks, which may have non-differentiable components and must avoid vanishing and exploding gradients for effective learning. However, since gradient descent is not applicable to hard-threshold functions, it is not clear how to learn networks of them in a principled way. We address this problem by observing that setting targets for hard-threshold hidden units in order to minimize loss is a discrete optimization problem, and can be solved as such. The discrete optimization goal is to find a set of targets such that each unit, including the output, has a linearly separable problem to solve. Given these targets, the network decomposes into individual perceptrons, which can then be learned with standard convex approaches. Based on this, we develop a recursive mini-batch algorithm for learning deep hard-threshold networks that includes the popular but poorly justified straight-through estimator as a special case. Empirically, we show that our algorithm improves classification accuracy in a number of settings, including for AlexNet and ResNet-18 on ImageNet, when compared to the straight-through estimator.

## 1 Introduction

The original approach to neural classification was to learn single-layer models with hard-threshold activations, like the perceptron (Rosenblatt, 1958). However, it proved difficult to extend these methods to multiple layers, because hard-threshold units, having zero derivative almost everywhere and being discontinuous at the origin, cannot be trained by gradient descent. Instead, the community turned to multilayer networks with soft activation functions, such as the sigmoid and, more recently, the ReLU, for which gradients can be computed efficiently by backpropagation (Rumelhart et al., 1986).

This approach has enjoyed remarkable success, enabling researchers to train networks with hundreds of layers and learn models that have significantly higher accuracy on a variety of tasks than any previous approach. However, as networks become deeper and wider, there has been a growing trend towards using hard-threshold activations for quantization purposes, where they enable binary or low-precision inference (e.g., Hubara et al. (2016); Rastegari et al. (2016); Zhou et al. (2016); Lin & Talathi (2016); Zhu et al. (2017)) and training (e.g., Lin et al. (2016); Li et al. (2017); Tang et al. (2017); Micikevicius et al. (2017)), which can greatly reduce the energy and computation time required by modern deep networks. Beyond quantization, the scale of the output of hard-threshold units is independent of (or insensitive to) the scale of their input, which can alleviate vanishing and exploding gradient issues and should help avoid some of the pathologies that occur during low-precision training with backpropagation (Li et al., 2017). Avoiding these issues is crucial for developing large systems of deep networks that can be used to perform even more complex tasks.

For these reasons, we are interested in developing well-motivated and efficient techniques for learning deep neural networks with hard-threshold units. In this work, we propose a framework for learning deep hard-threshold networks that stems from the observation that hard-threshold units output discrete values, indicating that combinatorial optimization may provide a principled method for training these networks. By specifying a set of discrete targets for each hidden-layer activation, the network

decomposes into many individual perceptrons, each of which can be trained easily given its inputs and targets. The difficulty in learning a deep hard-threshold network is thus in setting the targets so that each trained perceptron – including the output units – has a linearly separable problem to solve and thus can achieve its targets. We show that networks in which this is possible can be learned using our mixed convex-combinatorial optimization framework.

Building on this framework, we then develop a recursive algorithm, feasible target propagation (FTPROP), for learning deep hard-threshold networks. Since this is a discrete optimization problem, we develop heuristics for setting the targets based on per-layer loss functions. The mini-batch version of FTPROP can be used to explain and justify the oft-used straight-through estimator (Hinton, 2012; Bengio et al., 2013), which can now be seen as an instance of FTPROP with a specific choice of per-layer loss function and target heuristic. Finally, we develop a novel per-layer loss function that improves learning of deep hard-threshold networks. Empirically, we show improvements for our algorithm over the straight-through estimator on CIFAR-10 for two convolutional networks and on ImageNet for AlexNet and ResNet-18, with multiple types of hard-threshold activation.

RELATED WORK

The most common method for learning deep hard-threshold networks is to use backpropagation with the straight-through estimator (STE) (Hinton, 2012; Bengio et al., 2013), which simply replaces the derivative of each hard-threshold unit with the identity function. The STE is used in the quantized network literature (see citations above) to propagate gradients through quantized activations, and is used in Shalev-Shwartz et al. (2017) for training with flat activations. Later work generalized the STE to replace the hard-threshold derivative with other functions, including saturated versions of the identity function (Hubara et al., 2016). However, while the STE tends to work quite well in practice, we know of no rigorous justification or analysis of why it works or how to choose replacement derivatives. Beyond being unsatisfying in this regard, the STE is not well understood and can lead to gradient mismatch errors, which compound as the number of layers increases (Lin & Talathi, 2016). We show here that the STE, saturated STE, and all types of STE that we have seen are special cases of our framework, thus providing a principled justification for it and a basis for exploring and understanding alternatives.

Another common approach to training with hard-threshold units is to use randomness, either via stochastic neurons (e.g., Bengio et al. (2013); Hubara et al. (2016)) or probabilistic training methods, such as those of Soudry et al. (2014) or Williams (1992), both of which are methods for softening hard-threshold units. In contrast, our goal is to learn networks with deterministic hard-threshold units.

Finally, target propagation (TP) (LeCun, 1986; 1987; Carreira-Perpiñán & Wang, 2014; Bengio, 2014; Lee et al., 2015; Taylor et al., 2016) is a method that explicitly associates a target with the output of each activation in the network, and then updates each layer's weights to make its activations more similar to the targets. Our framework can be viewed as an instance of TP that uses combinatorial optimization to set discrete targets, whereas previous approaches employed continuous optimization to set continuous targets. The MADALINE Rule II algorithm (Winter & Widrow, 1988) can also be seen as a special case of our framework and of TP, where only one target is set at a time.

## 2 LEARNING DEEP NETWORKS WITH HARD-THRESHOLD UNITS

Given a dataset $\mathcal{D} = \{(\mathbf{x}^{(i)}, t^{(i)})\}_{i=1}^m$ with vector-valued inputs $\mathbf{x}^{(i)} \in \mathbb{R}^n$ and binary targets $t \in \{-1, +1\}$, we are interested in learning an $\ell$-layered deep neural network with hard-threshold units

$$y = f(\mathbf{x}; W) = g(W_\ell \, g(W_{\ell-1} \ldots g(W_1 \mathbf{x}) \ldots )), \tag{1}$$

with weight matrices $W = \{W_d : W_d \in \mathbb{R}^{n_d \times n_{d-1}}\}_{d=1}^\ell$ and element-wise activation function $g(\mathbf{x}) = \mathrm{sign}(\mathbf{x})$, where sign is the sign function such that $\mathrm{sign}(x) = 1$ if $x > 0$ and $-1$ otherwise. Each layer $d$ has $n_d$ units, where we define $n_0 = n$ for the input layer, and we let $\mathbf{h}_d = g(W_d \ldots g(W_1 \mathbf{x}) \ldots )$ denote the output of each hidden layer, where $\mathbf{h}_d = (h_{d1}, \ldots, h_{dn_d})$ and $h_{dj} \in \{-1, +1\}$ for each layer $d$ and each unit $j$. Similarly, we let $\mathbf{z}_d = W_d \, g(\ldots g(W_1 \mathbf{x}) \ldots )$ denote the pre-activation output of layer $d$. For compactness, we have incorporated the bias term into the weight matrices. We denote a row or column of a matrix $W_d$ as $W_{d,:j}$ and $W_{d,j:}$, respectively, and the entry in the $j$th row and $k$th column as $W_{d,jk}$. Using matrix notation, we can write this model as $Y = f(X; W) = g(W_\ell \ldots g(W_1 X) \ldots )$, where $X$ is the $n \times m$ matrix of dataset instances and $Y$ is the $n_\ell \times m$ matrix of outputs. We let $T_\ell$ denote the matrix of final-layer targets, $H_d$ denote the $n_d \times m$ matrix of hidden activations at layer $d$, and $Z_d$ denote the $n_d \times m$ matrix of pre-activations

Figure 1: After setting the hidden-layer targets $T_1$ of a deep hard-threshold network, the network decomposes into independent perceptrons, which can then be learned with standard methods.

at layer $d$. Our goal will be to learn $f$ by finding the weights $W$ that minimize an aggregate loss $L(Y, T_\ell) = \sum_{i=1}^m L(y^{(i)}, t^{(i)})$ for some convex per-instance loss $L(y, t)$.

In the simplest case, a hard-threshold network with no hidden layers is a perceptron $Y = g(W_1 X)$, as introduced by Rosenblatt (1958). The goal of learning a perceptron, or any hard-threshold network, is to classify unseen data. A useful first step is to be able to correctly classify the training data, which we focus on here for simplicity when developing our framework; however, standard generalization techniques such as regularization are easily incorporated into this framework and we do this for the experiments. Since a perceptron is a linear classifier, it is only able to separate a linearly-separable dataset.

**Definition 1.** *A dataset* $\{(\mathbf{x}^{(i)}, t^{(i)})\}_{i=1}^m$ *is* linearly separable *iff there exists a vector* $\mathbf{w} \in \mathbb{R}^n$ *and a real number* $\gamma > 0$ *such that* $(\mathbf{w} \cdot \mathbf{x}^{(i)}) t^{(i)} > \gamma$ *for all* $i = 1 \dots m$.

When a dataset is linearly separable, the perceptron algorithm is guaranteed to find its separating hyperplane in a finite number of steps (Novikoff, 1962), where the number of steps required is dependent on the size of the margin $\gamma$. However, linear separability is a very strong condition, and even simple functions, such as XOR, are not linearly separable and thus cannot be learned by a perceptron (Minsky & Papert, 1969). We would thus like to be able to learn multilayer hard-threshold networks.

Consider a simple single-hidden-layer hard-threshold network $Y = f(X; W) = g(W_2 \, g(W_1 X)) = g(W_1 H_1)$ for a dataset $\mathcal{D} = (X, T_2)$, where $H_1 = g(W_1 X)$ are the hidden-layer activations. An example of such a network is shown on the left side of Figure 1. Clearly, $Y$ and $H_1$ are both collections of (single-layer) perceptrons. Backpropagation cannot be used to train the input layer's weights $W_1$ because of the hard-threshold activations but, since each hidden activation $h_{1j}$ is the output of a perceptron, if we knew the value $t_{1j} \in \{-1, +1\}$ that each hidden unit *should* take for each input $\mathbf{x}^{(i)}$, we could then use the perceptron algorithm to set the first-layer weights, $W_1$, to produce these target values. We refer to $t_{1j}$ as the *target* of $h_{1j}$. Given a matrix of hidden-layer targets $T_1 \in \{-1, +1\}^{n_1 \times m}$, each layer (and in fact each perceptron in each layer) can be learned separately, as they no longer depend on each other, where the goal of perceptron learning is to update the weights of each layer $d$ so that its activations $H_d$ equal its targets $T_d$ given inputs $T_{d-1}$. Figure 1 shows an example of this decomposition. We denote the targets of an $\ell$-layer network as $T = \{T_1, \dots, T_\ell\}$, where $T_k$ for $k = 1 \dots \ell - 1$ are the hidden-layer targets and $T_\ell$ are the dataset targets. We often let $T_0 = X$ for notational convenience.

Auxiliary-variable-based approaches, such as ADMM (Taylor et al., 2016; Carreira-Perpiñán & Wang, 2014) and other target propagation methods (LeCun, 1986; Lee et al., 2015) use a similar process for decomposing the layers of a network; however, these focus on continuous variables and impose (soft) constraints to ensure that each activation equals its auxiliary variable. We take a different approach here, inspired by the combinatorial nature of the problem and the perceptron algorithm.

Since the final layer is a perceptron, the training instances can only be separated if the hidden-layer activations $H_1$ are linearly separable with respect to the dataset targets $T_2$. Thus, the hidden-layer targets $T_1$ must be set such that they are linearly separable with respect to the dataset targets $T_2$, since the hidden-layer targets $T_1$ are the intended values of their activations $H_1$. However, in order to ensure that the hidden-layer activations $H_1$ will equal their targets $T_1$ after training, the hidden-layer targets $T_1$ must be able to be produced (exactly) by the first layer, which is only possible if the hidden-layer targets $T_1$ are also linearly separable with respect to the inputs $X$. Thus, a sufficient condition for $f(X; W)$ to separate the data is that the hidden-layer targets induce linear separability in all units in both layers of the network. We refer to this property as *feasibility*.

**Definition 2.** *A setting of the targets* $T = \{T_1, \dots, T_\ell\}$ *of an* $\ell$-*layer deep hard-threshold network* $f(X; W)$ *is* feasible *for a dataset* $\mathcal{D} = (X, T_\ell)$ *iff for each unit* $j = 1 \dots n_d$ *in each layer* $d = 1 \dots \ell$ *the dataset formed by its inputs* $T_{d-1}$ *and targets* $T_{d,j:}$ *is linearly separable, where* $T_0 \triangleq X$.

Feasibility is a much weaker condition than linear separability, since the output decision boundary of a multilayer hard-threshold network with feasible targets is in general highly nonlinear. It follows from the definition of feasibility and convergence of the perceptron algorithm that if a feasible setting of a network's targets on a dataset exists, the network can separate the training data.

**Proposition 1.** *Let $\mathcal{D} = \{(\mathbf{x}^{(i)}, t^{(i)})\}$ be a dataset and let $f(X; W)$ be an $\ell$-layer hard-threshold network with feasible targets $T = \{T_1, \ldots, T_\ell\}$ in which each layer $d$ of $f$ was trained separately with inputs $T_{d-1}$ and targets $T_d$, where $T_0 \triangleq X$, then $f$ will correctly classify each instance $\mathbf{x}^{(i)}$, such that $f(\mathbf{x}^{(i)}; W)t^{(i)} > 0$ for all $i = 1 \ldots m$.*

Learning a deep hard-threshold network thus reduces to finding a feasible setting of its targets and then optimizing its weights given these targets, i.e., mixed convex-combinatorial optimization. The simplest method for this is to perform exhaustive search on the targets. Exhaustive search iterates through all possible settings of the hidden-layer targets, updating the weights of each perceptron whose inputs or targets changed, and returns the weights and feasible targets that result in the lowest loss. While impractical, exhaustive search is worth briefly examining to better understand the solution space. In particular, because of the decomposition afforded by setting the targets, exhaustive search over just the targets is sufficient to learn the globally optimal deep hard-threshold network, even though the weights are learned by gradient descent.

**Proposition 2.** *If a feasible setting of a deep hard-threshold network's targets on a dataset $\mathcal{D}$ exists, then exhaustive search returns the global minimum of the loss in time exponential in the number of hidden units.*

Learning can be improved and feasibility relaxed if, instead of the perceptron algorithm, a more robust method is used for perceptron learning. For example, a perceptron can be learned for a non-linearly-separable dataset by minimizing the hinge loss $L(z, t) = \max(0, 1 - tz)$, a convex loss on the perceptron's pre-activation output $z$ and target $t$ that maximizes the margin when combined with L2 regularization. In general, however, any method for learning linear classifiers can be used. We denote the loss used to train the weights of a layer $d$ as $L_d$, where the loss of the final layer $L_\ell$ is the output loss.

At the other end of the search spectrum is hill climbing. In each iteration, hill climbing evaluates all neighboring states of the current state (i.e., target settings that differ from the current one by only one target) and chooses the one with the lowest loss. The search halts when none of the new states improve the loss. Each state is evaluated by optimizing the weights of each perceptron given the state's targets, and then computing the output loss. Hill climbing is more practical than exhaustive search, since it need not explore an exponential number of states, and it also provides the same local optima guarantee as gradient descent on soft-threshold networks.

**Proposition 3.** *Hill climbing on the targets of a deep hard-threshold network returns a local minimum of the loss, where each iteration takes time linear in the size of the set of proposed targets.*

Exhaustive search and hill climbing comprise two ends of the discrete optimization spectrum. Beam search, which maintains a beam of the most promising solutions and explores each, is another powerful approach that contains both hill climbing and exhaustive search as special cases. In general, however, any discrete optimization algorithm can be used for setting targets. For example, methods from satisfiability solving, integer linear programming, or constraint satisfaction might work well, as the linear separability requirements of feasibility can be viewed as constraints on the search space.

We believe that our mixed convex-combinatorial optimization framework opens many new avenues for developing learning algorithms for deep networks, including those with non-differentiable modules. In the following section, we use these ideas to develop a learning algorithm that hews much closer to standard methods, and in fact contains the straight-through estimator as a special case.

## 3   FEASIBLE TARGET PROPAGATION

The open question from the preceding section is how to set the hidden-layer targets. Generating good, feasible targets for the entire network at once is a difficult problem; instead, an easier approach is to propose targets for only one layer at a time. As in backpropagation, it makes sense to start from the output layer, since the final-layer targets are given, and successively set targets for each upstream layer. Further, since it is hard to know a priori if a setting of a layer's targets is feasible for a given network architecture, a simple alternative is to set the targets for a layer $d$ and then optimize the upstream weights (i.e., weights in layers $j \leq d$) to check if the targets are feasible. Since the goals

when optimizing a layer's weights and when setting its upstream targets (i.e., its inputs) are the same – namely, to induce feasibility – a natural method for setting target values is to choose targets that reduce the layer's loss $L_d$. However, because the targets are discrete, moves in target space are large and non-smooth and cannot be guaranteed to lower the loss without actually performing the move. Thus, heuristics are necessary. We discuss these in more detail below.

Determining feasibility of the targets at layer $d$ can be done by recursively updating the weights of layer $d$ and proposing targets for layer $d - 1$ given the targets for layer $d$. This recursion continues until the input layer is reached, where feasibility (i.e., linear separability) can be easily determined by optimizing that layer's weights given its targets and the dataset inputs. The targets at layer $d$ can then be updated based on the information gained from the recursion and, if the upstream weights were altered, based on the new outputs of layer $d - 1$. We call this recursive algorithm *feasible target propagation*, or FTPROP. Pseudocode is shown in Algorithm 1.

---

**Algorithm 1** Train an $\ell$-layer hard-threshold network $Y = f(X; W)$ on dataset $\mathcal{D} = (X, T_\ell)$ with feasible target propagation (FTPROP) using loss functions $L = \{L_d\}_{d=1}^{\ell}$.

1: initialize weights $W = \{W_1, \ldots, W_\ell\}$ randomly
2: initialize targets $T_1, \ldots, T_{\ell-1}$ as the outputs of their hidden units in $f(X; W)$
3: set $T_0 \leftarrow X$ and set $T \leftarrow \{T_0, T_1, \ldots, T_\ell\}$
4: FTPROP($W, T, L, \ell$)         *// train the network by searching for a feasible target setting*

5: **function** FTPROP(weights $W$, targets $T$, losses $L$, and layer index $d$)
6:     optimize $W_d$ with respect to layer loss $L_d(Z_d, T_d)$       *// check feasibility; $Z_d = W_d T_{d-1}$*
7:     **if** activations $H_d = g(W_d T_{d-1})$ equal the targets $T_d$ **then return** True      *// feasible*
8:     **else if** this is the first layer (i.e., $d = 1$) **then return** False      *// infeasible*
9:     **while** computational budget of this layer not exceeded **do**     *// e.g., determined by beam search*
10:        $T_{d-1} \leftarrow$ heuristically set targets for upstream layer to reduce layer loss $L_d(Z_d, T_d)$
11:        **if** FTPROP($W, T, L, d-1$) **then**       *// check if targets $T_{d-1}$ are feasible*
12:           optimize $W_d$ with respect to layer loss $L_d(Z_d, T_d)$
13:           **if** activations $H_d = g(W_d T_{d-1})$ equal the targets $T_d$ **then return** True      *// feasible*
14:     **return** False

---

As the name implies, FTPROP is a form of target propagation (LeCun, 1986; 1987; Lee et al., 2015) that uses discrete optimization to set discrete targets, instead of using continuous optimization to set continuous targets. FTPROP is also highly related to RDIS (Friesen & Domingos, 2015), a powerful nonconvex optimization algorithm based on satisfiability (SAT) solvers that recursively chooses and sets subsets of variables in order to decompose the underlying problem into simpler subproblems. While RDIS is applied only to continuous problems, the ideas behind RDIS can be generalized to discrete variables via the sum-product theorem (Friesen & Domingos, 2016). This suggests an interesting connection between FTPROP and SAT that we leave for future work.

Of course, modern deep networks will not always have a feasible setting of their targets for a given dataset. For example, a convolutional layer imposes a large amount of structure on its weight matrix, making it less likely that the layer's input will be linearly separable with respect to its targets. Further, ensuring feasibility will in general cause learning to overfit the training data, which will worsen generalization performance. Thus, we would like to relax the feasibility requirements.

In addition, there are many benefits of using mini-batch instead of full-batch training, including improved generalization gap (e.g., see LeCun et al. (2012) or Keskar et al. (2016)), reduced memory usage, the ability to exploit data augmentation, and the prevalence of tools (e.g., GPUs) designed for it.

Fortunately, it is straightforward to convert FTPROP to a mini-batch algorithm and to relax the feasibility requirements. In particular, since it is important not to overcommit to any one mini-batch, the mini-batch version of FTPROP (i) only updates the weights and targets of each layer once per mini-batch; (ii) only takes a small gradient step on each layer's weights, instead of optimizing them fully; (iii) sets the targets of the downstream layer in parallel with updating the current layer's weights, since the weights will not change much; and (iv) removes all checks for feasibility. We call this algorithm FTPROP-MB and present pseudocode in Algorithm 2. FTPROP-MB closely resembles backpropagation-based methods, allowing us to easily implement it with standard libraries.

---

**Algorithm 2** Train an $\ell$-layer hard-threshold network $Y = f(X; W)$ on dataset $\mathcal{D} = (X, T_\ell)$ with mini-batch feasible target propagation (FTPROP-MB) using loss functions $L = \{L_d\}_{d=1}^{\ell}$.

1: initialize weights $W = \{W_1, \dots, W_\ell\}$ randomly
2: **for** each minibatch $(X_b, T_b)$ from $\mathcal{D}$ **do**
3:    initialize targets $T_1, \dots, T_{\ell-1}$ as the outputs of their hidden units in $f(X_b; W)$ // *forward pass*
4:    set $T_0 \leftarrow X_b$, set $T_\ell \leftarrow T_b$, and set $T \leftarrow \{T_0, \dots, T_\ell\}$
5:    FTPROP-MB($W, T, L, \ell$)

6: **function** FTPROP-MB(weights $W$, targets $T$, losses $L$, and layer index $d$)
7:    $\hat{T}_{d-1} \leftarrow$ set targets for upstream layer based on current weights $W_d$ and loss $L_d(Z_d, T_d)$
8:    update $W_d$ with respect to layer loss $L_d(Z_d, T_d)$         // *where $Z_d = W_d T_{d-1} = W_d H_{d-1}$*
9:    **if** $d > 1$ **then** FTPROP-MB($W, \{T_0, \dots, \hat{T}_{d-1}, \dots, T_\ell\}, L, d-1$)

---

## 3.1 TARGET HEURISTICS

When the activations of each layer are differentiable, backpropagation provides a method for telling each layer how to adjust its outputs to improve the loss. Conversely, in hard-threshold networks, target propagation provides a method for telling each layer how to adjust its outputs to improve the next layer's loss. While gradients cannot propagate through hard-threshold units, the derivatives within a layer can still be computed. An effective and efficient heuristic for setting the target $t_{dj}$ for an activation $h_{dj}$ of layer $d$ is to use the (negative) sign of the partial derivative of the next layer's loss. Specifically, we set $t_{dj} = r(h_{dj})$, where

$$r(h_{dj}) \triangleq \text{sign}\left(-\frac{\partial}{\partial h_{dj}} L_{d+1}(Z_{d+1}, T_{d+1})\right) \tag{2}$$

and $Z_{d+1}$ is either the pre-activation or post-activation output, depending on the choice of loss.

When used to update only a single target at a time, this heuristic will often set the target value that correctly results in the lowest loss. In particular, when $L_{d+1}$ is convex, its negative partial derivative with respect to $h_{dj}$ by definition points in the direction of the global minimum of $L_{d+1}$. Without loss of generality, let $h_{dj} = -1$. Now, if $r(h_{dj}) = -1$, then it follows from the convexity of the loss that flipping $h_{dj}$ and keeping all other variables the same would increase $L_{d+1}$. On the other hand, if $r(h_{dj}) = +1$, then flipping $h_{dj}$ may or may not reduce the loss, since convexity cannot tell us which of $h_{dj} = +1$ or $h_{dj} = -1$ results in a smaller $L_{d+1}$. However, the discrepancy between $h_{dj}$ and $r(h_{dj})$ indicates a lack of confidence in the current value of $h_{dj}$. A natural choice is thus to set $t_{dj}$ to push the pre-activation value of $h_{dj}$ towards 0, making $h_{dj}$ more likely to flip. Setting $t_{dj} = r(h_{dj}) = +1$ accomplishes this. We note that, while this heuristic performs well, there is still room for improvement, for example by extending $r(\cdot)$ to better handle the $h_{dj} \neq r(h_{dj})$ case or by combining information across the batch. We leave such investigations for future work.

## 3.2 LAYER LOSS FUNCTIONS

The hinge loss, shown in Figure 2a, is a robust version of the perceptron criterion and is thus a natural per-layer loss function to use for finding good settings of the targets and weights, even when there are no feasible target settings. However, in preliminary experiments we found that learning tended to stall and become erratic over time when using the hinge loss for each layer. We attribute this to two separate issues. First, the hinge loss is sensitive to noisy data and outliers (Wu & Liu, 2007), which can cause learning to focus on instances that are unlikely to ever be classified correctly, instead of on instances near the separator. Second, since with convolutional layers and large, noisy datasets it is unlikely that a layer's inputs are entirely linearly separable, it is important to prioritize some targets over others. Ideally, the highest priority targets would be those with the largest effect on the output loss.

The first issue can be solved by saturating (truncating) the hinge loss, thus making it less sensitive to outliers (Wu & Liu, 2007). The saturated hinge loss, shown in Figure 2b, is $\text{sat\_hinge}(z, t; b) = \max(0, 1 - \max(tz, b))$ for some threshold $b$, where we set $b = -1$ to make its derivative symmetric. The second problem can be solved in a variety of ways, including randomly subsampling targets or weighting the loss associated with each target according to some heuristic. The simplest and most accurate method that we have found is to weight the loss for each target $t_{dj}$ by the magnitude of the

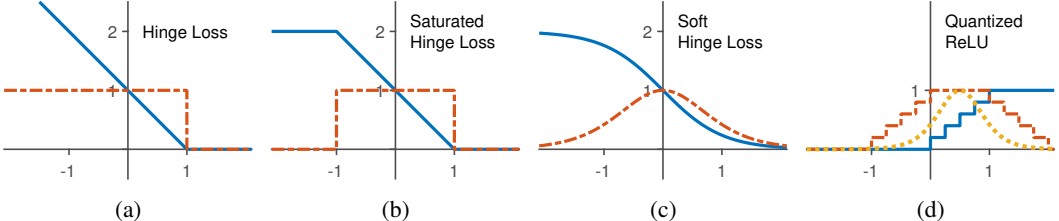

Figure 2: Figures (a)-(c) show different per-layer loss functions (solid blue line) and their derivatives (dashed red line). Figure (d) shows the quantized ReLU activation (solid blue line), which is a sum of step functions, its corresponding sum of saturated-hinge-loss derivatives (dashed red line), and the soft-hinge-loss approximation to this sum that was found to work best (dotted yellow line).

partial derivative of the next layer's loss $L_{d+1}$ with respect to the target's hidden unit $h_{dj}$, such that

$$L_d(z_{dj}, t_{dj}) = \text{sat\_hinge}(z_{dj}, t_{dj}) \cdot \left| \frac{\partial L_{d+1}}{\partial h_{dj}} \right|. \tag{3}$$

While the saturated hinge loss works well, if the input $z_{dj}$ ever moves out of the range $[-1, +1]$ then its derivative will become zero and the unit will no longer be trainable. To avoid this, we propose the *soft hinge loss*, shown in Figure 2c, where $\text{soft\_hinge}(z, t) = \tanh(-tz) + 1$. Like the saturated hinge, the soft hinge has slope 1 at the threshold and has a symmetric derivative; however, it also benefits from having a larger input region with non-zero derivative. Note that Bengio et al. (2013) report that using the derivative of a sigmoid as the STE performed worse than the identity function. Based on our experiments with other loss functions, including variations of the squared hinge loss and the log loss, this is most likely because the slope of the sigmoid is less than unity at the threshold, which causes vanishing gradients. Loss functions with asymmetric derivatives around the threshold also seemed to perform worse than those with symmetric derivatives (e.g., the saturating and soft hinge losses). In our experiments, we show that the soft hinge loss outperforms the saturated hinge loss for both sign and quantized ReLU activations, which we discuss below.

### 3.3 RELATIONSHIP TO THE STRAIGHT-THROUGH ESTIMATOR

When each loss term in each hidden layer is scaled by the magnitude of the partial derivative of its downstream layer's loss and each target is set based on the sign of the same partial derivative, then target propagation transmits information about the output loss to every layer in the network, despite the hard-threshold units. Interestingly, this combination of loss function and target heuristic can exactly reproduce the weight updates of the straight-through estimator (STE). Specifically, the weight updates that result from using the scaled saturated hinge loss from (3) and the target heuristic in (2) are exactly those of the saturated straight-through estimator (SSTE) defined in Hubara et al. (2016), which replaces the derivative of $\text{sign}(z)$ with $1_{|z| \leq 1}$, where $1_{(\cdot)}$ is the indicator function. Other STEs correspond to different choices of per-layer loss function. For example, the original STE corresponds to the linear loss $L(z, t) = -tz$ with the above target heuristic. This connection provides a justification for existing STE approaches, which can now each be seen as an instance of FTPROP with a particular choice of per-layer loss function and target heuristic. We believe that this will enable more principled investigations and extensions of these methods in future work.

### 3.4 QUANTIZED ACTIVATIONS

Straight-through estimation is also commonly used to backpropagate through quantized variants of standard activations, such as the ReLU. Figure 2d shows a quantized ReLU (qReLU) with 6 evenly-spaced quantization levels. The simplest and most popular straight-through estimator (STE) for qReLU is to use the derivative of the saturated (or clipped) ReLU $\frac{\partial \text{sat\_ReLU}(x)}{\partial x} = 1_{0 < x < 1}$, where $\text{sat\_ReLU}(x) = \min(1, \max(x, 0))$. However, if we instead consider the qReLU activation from the viewpoint of FTPROP, then the qReLU becomes a (normalized) sum of step functions $\text{qReLU}(z) = \frac{1}{k} \sum_{i=0}^{k-1} \text{step}(z - \frac{i}{k-1})$, where $\text{step}(z) = 1$ if $z > 0$ and 0 otherwise, and is a linear transformation of $\text{sign}(z)$. The resulting derivative of the sum of saturated hinge losses (one for each step function) is shown in red in Figure 2d, and is clearly quite different than the STE described above. In initial experiments, this performed as well as or better than the STE; however, we achieved additional performance improvements by using the softened approximation shown in yellow in Figure 2d, which is simply the derivative of a soft hinge that has been scaled and shifted to match the

Table 1: The best top-1 test accuracy for each network over all epochs when trained with sign, qReLU, and full-precision baseline activations on CIFAR-10 and ImageNet. The hard-threshold activations are trained with both FTPROP-MB with per-layer soft hinge losses (FTP-SH) and the saturated straight-through estimator (SSTE). Bold numbers denote the best performing quantized activation in each experiment.

|  | Sign | | qReLU | | Baselines | |
| --- | --- | --- | --- | --- | --- | --- |
|  | *SSTE* | *FTP-SH* | *SSTE* | *FTP-SH* | *ReLU* | *Sat. ReLU* |
| 4-layer convnet (CIFAR-10) | 80.6 | **81.3** | 85.6 | 85.5 | 86.5 | 87.3 |
| 8-layer convnet (CIFAR-10) | 84.6 | **84.9** | 88.4 | **89.8** | 91.2 | 91.2 |
| AlexNet (ImageNet) | 46.7 | **47.3** | 59.4 | **60.7** | 61.3 | 61.9 |
| ResNet-18 (ImageNet) | **49.1** | 47.8 | 60.6 | **64.3** | 69.1 | 66.9 |

qReLU domain. This is a natural choice because the derivative of a sum of a small number of soft hinge losses has a shape similar to that of the derivative of a single soft hinge loss.

## 4 EXPERIMENTS

We evaluated FTPROP-MB with soft hinge per-layer losses (FTP-SH) for training deep networks with sign and 2- and 3-bit qReLU activations by comparing models trained with FTP-SH to those trained with the saturated straight-through estimators (SSTEs) described earlier (although, as discussed, these SSTEs can also be seen as instances of FTPROP-MB). We compared to these SSTEs because they are the standard approach in the literature and they significantly outperformed the STE in our initial experiments (Hubara et al. (2016) observed similar behavior). Computationally, FTPROP-MB has the same performance as straight-through estimation; however, the soft hinge loss involves computing a hyperbolic tangent, which requires more computation than a piecewise linear function. This is the same performance difference seen when using sigmoid activations instead of ReLUs in soft-threshold networks. We also trained each model with ReLU and saturated-ReLU activations as full-precision baselines.

We did not use weight quantization because our main interest is training with hard-threshold activations, and because recent work has shown that weights can be quantized with little effect on performance (Hubara et al., 2016; Rastegari et al., 2016; Zhou et al., 2016). We tested these training methods on the CIFAR-10 (Krizhevsky, 2009) and ImageNet (ILSVRC 2012) (Russakovsky et al., 2015) datasets. On CIFAR-10, we trained a simple 4-layer convolutional network and the 8-layer convolutional network of Zhou et al. (2016). On ImageNet, we trained AlexNet (Krizhevsky et al., 2012), the most common model in the quantization literature, and ResNet-18 (He et al., 2015a). Further experiment details are provided in Appendix A, along with learning curves for all experiments, and code is available at `https://github.com/afriesen/ftprop`.

### 4.1 CIFAR-10

Test accuracies for the 4-layer and 8-layer convolutional networks on CIFAR-10 are shown in Table 1. For the 4-layer model, FTP-SH shows a consistent 0.5-1% accuracy gain over SSTE for the entire training trajectory, resulting in the 0.7% improvement shown in Table 1. However, for the 2-bit qRELU activation, SSTE and FTP-SH perform nearly identically in the 4-layer model. Conversely, for the more complex 8-layer model, the FTP-SH accuracy is only 0.3% above SSTE for the sign activation, but for the qReLU activation FTP-SH achieves a consistent 1.4% improvement over SSTE.

We posit that the decrease in performance gap for the sign activation when moving from the 4- to 8-layer model is because both methods are able to effectively train the higher-capacity model to achieve close to its best possible performance on this dataset, whereas the opposite is true for the qReLU activation; i.e., the restricted capacity of the 4-layer model limits the ability of both methods to train the more expressive qReLU effectively. If this is true, then we expect that FTP-SH will outperform SSTE for both the sign and qReLU activations on a harder dataset. Unsurprisingly, none of the low-precision methods perform as well as the baseline high-precision methods; however, the narrowness of the performance gap between 2-bit qReLU with FTP-SH and full-precision ReLU is encouraging.

### 4.2 IMAGENET

The results from the ImageNet experiments are also shown in Table 1. As predicted from the CIFAR-10 experiments, we see that FTP-SH improves test accuracy on AlexNet for both sign and 2-bit

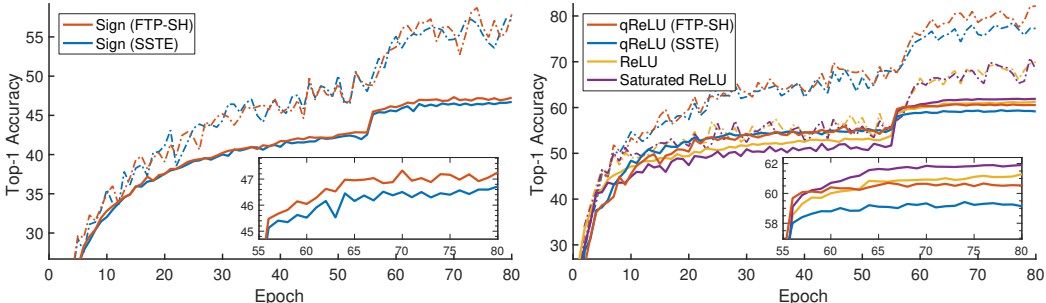

Figure 3: The top-1 train (thin dashed lines) and test (thicker solid lines) accuracies for AlexNet with different activation functions on ImageNet. The inset figures show the test accuracy for the final 25 epochs in detail. In both figures, FTPROP-MB with soft hinge (FTP-SH, red) outperforms the saturated straight-through estimator (SSTE, blue). The left figure shows the network with sign activations. The right figure shows that the 2-bit quantized ReLU (qReLU) trained with our method (FTP-SH) performs nearly as well as the full-precision ReLU. Interestingly, saturated ReLU outperforms standard ReLU. Best viewed in color.

qReLU activations on the more challenging ImageNet dataset. This is also shown in Figure 3, which plots the top-1 train and test accuracy curves for the six different activation functions for AlexNet on ImageNet. The left-hand plot shows that training sign activations with FTP-SH provides consistently better test accuracy than SSTE throughout the training trajectory, despite the hyperparameters being optimized for SSTE. This improvement is even larger for the 2-bit qReLU activation in the right-hand plot, where the FTP-SH qReLU even outperforms the full-precision ReLU for part of its trajectory, and outperforms the SSTE-trained qReLU by almost 2%. Interestingly, we find that the saturated ReLU outperforms the standard ReLU by almost a full point of accuracy. We believe that this is due to the regularization effect caused by saturating the activation. This may also account for the surprisingly good performance of the FTP-SH qReLU relative to full-precision ReLU, as hard-threshold activations also provide a strong regularization effect.

Finally, we ran a single experiment with ResNet-18 on ImageNet, using hyperparameters from previous works that used SSTE, to check (i) whether the soft hinge loss exhibits vanishing gradient behavior due to its diminishing slope away from the origin, and (ii) to evaluate the performance of FTP-SH for a less-quantized ReLU (we used $k = 5$ steps, which is less than the full range of a 3-bit ReLU). While FTP-SH does slightly worse than SSTE for the sign function, we believe that this is because the hyperparameters were tuned for SSTE and not due to vanishing gradients, as we would expect much worse accuracy in that case. Results from the qReLU activation provide further evidence against vanishing gradients as FTP-SH for qReLU outperforms SSTE by almost 4% in top-1 accuracy (Table 1).

## 5 CONCLUSION

In this work, we presented a novel mixed convex-combinatorial optimization framework for learning deep neural networks with hard-threshold units. Combinatorial optimization is used to set discrete targets for the hard-threshold hidden units, such that each unit only has a linearly-separable problem to solve. The network then decomposes into individual perceptrons, which can be learned with standard convex approaches, given these targets. Based on this, we developed a recursive algorithm for learning deep hard-threshold networks, which we call feasible target propagation (FTPROP), and an efficient mini-batch variant (FTPROP-MB). We showed that the commonly-used but poorly-justified saturating straight-through estimator (STE) is the special case of FTPROP-MB that results from using a saturated hinge loss at each layer and our target heuristic and other types of STE correspond to other heuristic and loss combinations in FTPROP-MB. Finally, we defined the soft hinge loss and showed that FTPROP-MB with a soft hinge loss at each layer improves classification accuracy for multiple models on CIFAR-10 and ImageNet when compared to the saturating STE.

In future work, we plan to develop novel target heuristics and layer loss functions by investigating connections between our framework and constraint satisfaction and satisfiability. We also intend to further explore the benefits of deep networks with hard-threshold units. In particular, while recent research clearly shows their ability to reduce computation and energy requirements, they should also be less susceptible to vanishing and exploding gradients and may be less susceptible to covariate shift and adversarial examples.

ACKNOWLEDGMENTS

This research was partly funded by ONR grant N00014-16-1-2697. The GPU machine used for this research was donated by NVIDIA.

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

## A    EXPERIMENT DETAILS

All experiments were performed using PyTorch (`http://pytorch.org/`). CIFAR-10 experiments with the 4-layer convolutional network were performed on an NVIDIA Titan X. All other experiments were performed on NVIDIA Tesla P100 devices in a DGX-1. Code for the experiments is available at `https://github.com/afriesen/ftprop`.

### A.1    CIFAR-10

On CIFAR-10, which has 50K training images and 10K test images divided into 10 classes, we trained both a simple 4-layer convolutional network and a deeper 8-layer convolutional network used in (Zhou et al., 2016) with the above methods and then compared their top-1 accuracies on the test set. We pre-processed the images with mean / std normalization, and augmented the dataset with random horizontal flips and random crops from images padded with 4 pixels. Hyperparameters were chosen based on a small amount of exploration on a validation set.

The first network we tested on CIFAR-10 was a simple 4-layer convolutional network (convnet) structured as: conv(32) → conv(64) → fc(1024) → fc(10), where conv($c$) and fc($c$) indicate a convolutional layer and fully-connected layer, respectively, with $c$ channels. Both convolutional layers used $5 \times 5$ kernels. Max-pooling with stride 2 was used after each convolutional layer, and a non-linearity was placed before each of the above layers except the first. Adam (Kingma & Ba, 2015) with learning rate 2.5e-4 and weight decay 5e-4 was used to minimize the cross-entropy loss for 300 epochs. The learning rate was decayed by a factor of 0.1 after 200 and 250 epochs.

In order to evaluate the performance of FTPROP-MB with the soft hinge loss on a deeper network, we adapted the 8-layer convnet from Zhou et al. (2016) to CIFAR-10. This network has 7 convolutional layers and one fully-connected layer for the output and uses batch normalization (Ioffe & Szegedy, 2015) before each non-linearity. We optimized the cross-entropy loss with Adam using a learning rate of 1e-3 and a weight decay of 1e-7 for the sign activation and 5e-4 for the qReLU and baseline activations. We trained for 300 epochs, decaying the learning rate by 0.1 after 200 and 250 epochs.

### A.2    LEARNING CURVES FOR CIFAR-10

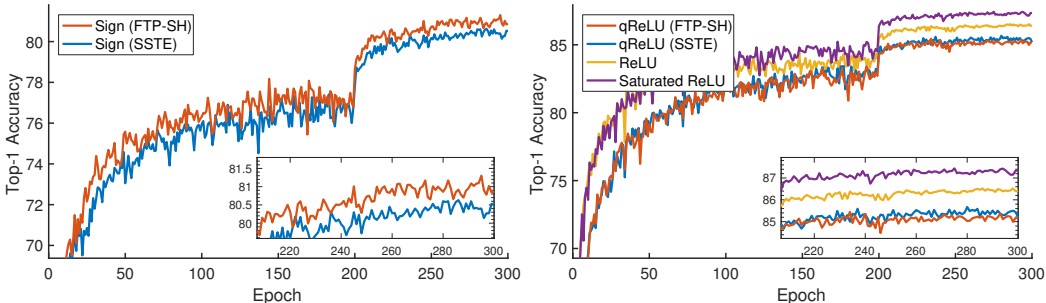

Figure 4: The top-1 test accuracies for the 4-layer convolutional network with different activation functions on CIFAR-10. The inset figures show the test accuracy for the final 100 epochs in detail. The left figure shows the network with sign activations. The right figure shows the network with 2-bit quantized ReLU (qReLU) activations and with the full-precision baselines. Best viewed in color.

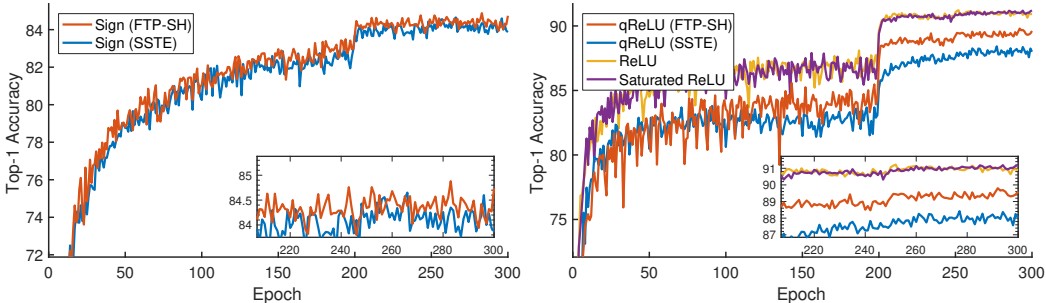

Figure 5: The top-1 test accuracies for the 8-layer convolutional network with different activation functions on CIFAR-10. The inset figures show the test accuracy for the final 100 epochs in detail. The left figure shows the network with sign activations. The right figure shows the network with 2-bit quantized ReLU (qReLU) activations and with the full-precision baselines. Best viewed in color.

## A.3 IMAGENET (ILSVRC 2012)

On ImageNet, a much more challenging dataset with roughly 1.2M training images and 50K validation images divided into 1000 classes, we trained AlexNet, the most commonly used model in the quantization literature, with different activations and compared top-1 and top-5 accuracies of the trained models on the validation set. As is standard practice, we treat the validation set as the test data. Images were resized to $256 \times 256$, mean / std normalized, and then randomly cropped to $224 \times 224$ and randomly horizontally flipped. Models are tested on centered $224 \times 224$ crops of the test images. Hyperparameters were set based on Zhou et al. (2016) and Zhu et al. (2017), which both used SSTE to train AlexNet on ImageNet.

We trained the Zhou et al. (2016) variant of AlexNet (Krizhevsky et al., 2012) on ImageNet with sign, 2-bit qReLU, ReLU, and saturated ReLU activations. This version of AlexNet removes the dropout and replaces the local contrast normalization layers with batch normalization. Our implementation does not split the convolutions into two separate blocks. We used the Adam optimizer with learning rate 1e-4 on the cross-entropy loss for 80 epochs, decaying the learning rate by 0.1 after 56 and 64 epochs. For the sign activation, we used a weight decay of 5e-6 as in Zhou et al. (2016). For the ReLU and saturated ReLU activations, which are much more likely to overfit, we used a weight decay of 5e-4, as used in Krizhevsky et al. (2012). For the 2-bit qReLU activation, we used a weight decay of 5e-5, since it is more expressive than sign but less so than ReLU.

As with AlexNet, we trained ResNet-18 (He et al., 2015b) on ImageNet with sign, qReLU, ReLU, and saturated ReLU activations; however, for ResNet-18 we used a qReLU with $k = 5$ steps (i.e., 6 quantization levels, requiring 3 bits). We used the ResNet code provided by PyTorch. We optimized the cross-entropy loss with SGD with learning rate 0.1 and momentum 0.9 for 90 epochs, decaying the learning rate by a factor of 0.1 after 30 and 60 epochs. For the sign activation, we used a weight decay of 5e−7. For the ReLU and saturated ReLU activations, we used a weight decay of 1e-4. For the qReLU activation, we used a weight decay of 1e-5.

## A.4 LEARNING CURVES FOR IMAGENET

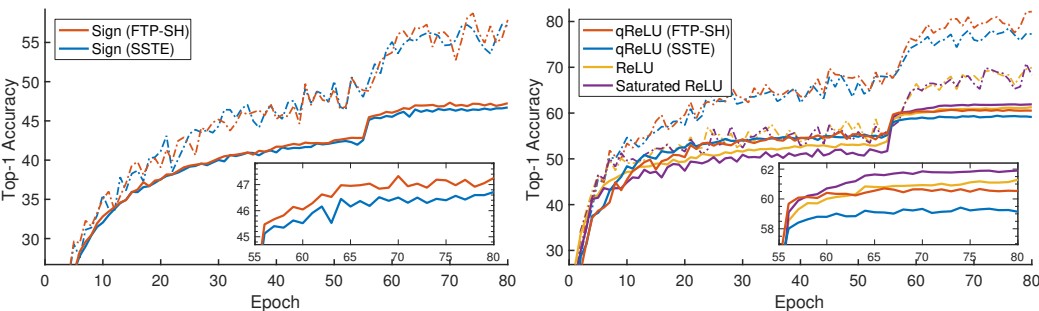

Figure 6: The top-1 train (thin dashed lines) and test (thicker solid lines) accuracies for AlexNet with different activation functions on ImageNet. The inset figures show the test accuracy for the final 25 epochs in detail. The left figure shows the network with sign activations. The right figure shows the network with 2-bit quantized ReLU (qReLU) activations and with the full-precision baselines. Best viewed in color.

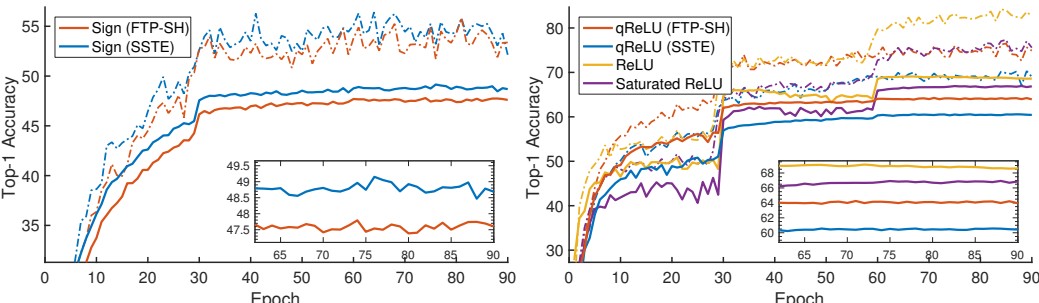

Figure 7: The top-1 train (thin dashed lines) and test (thicker solid lines) accuracies for ResNet-18 with different activation functions on ImageNet. The inset figures show the test accuracy for the final 60 epochs in detail. The left figure shows the network with sign activations. The right figure shows the network with 3-bit quantized ReLU (qReLU) activations and with the full-precision baselines. Best viewed in color.

