# OpenReview forum: "Deep Learning as a Mixed Convex-Combinatorial Optimization Problem"
_ICLR.cc/2018/Conference — Accept (Poster)_

### Official Review · AnonReviewer1 · 2017-11-26
**An interesting way of explaining and generalizing approaches for learning neural nets with hard activation**

**Rating:** 7
**Confidence:** 4

**Review:**

The paper studies learning in deep neural networks with hard activation functions, e.g. step functions like sign(x). Of course, backpropagation is difficult to adapt to such networks, so prior work has considered different approaches. Arguably the most popular is straight-through estimation (Hinton 2012, Bengio et al. 2013), in which the activation functions are simply treated as identity functions during backpropagation. More recently, a new type of straight-through estimation, saturated STE (Hubara et al., 2016) uses 1[|z|<1] as the derivative of sign(z).

The paper generalizes saturated STE by recognizing that other discrete targets of each activation layer can be chosen. Deciding on these targets is formulated as a combinatorial optimization problem. Once the targets are chosen, updating the weights of each layer to minimize the loss on those targets is a convex optimization. The targets are heuristically updated through the layers, starting out the output using the proposed feasibility target propagation. At each layer, the targets can be chosen using a variety of search algorithms such as beam search.

Experiments show that FTP often outperforms saturated STE on CIFAR and ImageNet with sign and quantized activation functions, reaching levels of performance closer to the full-precision activation networks.

This paper's ideas are very interesting, exploring an alternative training method to backpropagation that supports hard-threshold activation functions. The experimental results are encouraging, though I have a few questions below that prevent me for now from rating the paper higher.

Comments and questions:

1) How computationally expensive is FTP? The experiments using ResNet indicate it is not prohibitively expensive, but I am eager for more details.

2) Does (Hubara et al., 2016) actually compare their proposed saturated STE with the orignal STE on any tasks? I do not see a comparison. If that is so, should this paper also compare with STE? How do we know if generalizing saturated STE is more worthwhile than generalizing STE?

3) It took me a while to understand the authors' subtle comparison with target propagation, where they say "Our framework can be viewed as an instance of target propagation that uses combinatorial optimization to set discrete targets, whereas previous approaches employed continuous optimization." It seems that the difference is greater than explicitly stated, that prior target propagation used continuous optimization to set *continuous targets*. (One could imagine using continuous optimization to set discrete targets such as a convex relaxation of a constraint satisfaction problem.) Focusing on discrete targets gains the benefits of quantized networks. If I am understanding the novelty correctly, it would strengthen the paper to make this difference clear.

4) On a related note, if feasible target propagation generalizes saturated straight through estimation, is there a connection between (continuous) target propagation and the original type of straight through estimation?

5) In Table 1, the significance of the last two columns is unclear. It seems that ReLU and Saturated ReLU are included to show the performance of networks with full-precision activation functions (which is good). I am unclear though on why they are compared against each other (bolding one or the other) and if there is some correspondence between those two columns and the other pairs, i.e., is ReLU some kind of analog of SSTE and Saturated ReLU corresponds to FTP-SH somehow?

---

> ### Author Response · Authors · 2017-12-19
> **Response to Reviewer 1**
>
> Thank you for your review. We respond to each of your questions below.
>
> 1) FTP-SH is no more expensive than backprop (in the same way that SSTE isn’t either, and SSTE is a special case of FTPROP-MB). The only added cost is that the soft hinge loss requires computing an exponential, which is slower than a max (i.e., the cost of computing a sigmoid vs. a ReLU), but this is a minor difference in compute time.
>
> 2) In the experiments, Hubara et al. (2016) does not compare SSTE and STE directly, but in the text of the paper they report that “Not [saturating] the gradient when [the input] is too large significantly worsens performance.” This is also what we found in preliminary experiments, where the unsaturated STE is significantly worse than STE. Note, however, that STE is also a special case of our framework where the loss function is just loss(z, t) = -zt, so we generalize that as well (and pretty much any type of STE can be obtained by choosing different losses in our framework).
>
> 3) Yes, this is a good point and correct. We will update the paper to make this fact more clear. Thank you.
>
> 4) It’s possible, although if so it’s not an obvious connection, and we haven’t studied this issue in detail yet.
>
> 5) Yes, good point. This is somewhat confusing, and we will clarify it in the paper and remove the bolding, since the goal isn’t really to compare them against each other (although it is mildly interesting that saturating the ReLU improves performance in some cases). There is no correspondence between those two columns and the other pairs; the formatting of the table is just unclear.

---

### Official Review · AnonReviewer2 · 2017-11-27
**Well-organized analysis of the hard-threshold networks**

**Rating:** 7
**Confidence:** 4

**Review:**

This paper examines the problem of optimizing deep networks of hard-threshold units. This is a significant topic with implications for quantization for computational efficiency, as well as for exploring the space of learning algorithms for deep networks. While none of the contributions are especially novel, the analysis is clear and well-organized, and the authors do a nice job in connecting their analysis to other work.

---

### Official Review · AnonReviewer3 · 2017-12-02
**Nice discussion and solution to optimizing neural networks with hard thresholds but with some flaws**

**Rating:** 7
**Confidence:** 3

**Review:**

The paper discusses the problem of optimizing neural networks with hard threshold and proposes a novel solution to it. The problem is of significance because in many applications one requires deep networks which uses reduced computation and limited energy. The authors frame the problem of optimizing such networks to fit the training data as a convex combinatorial problems. However since the complexity of such a problem is exponential, the authors propose a collection of heuristics/approximations to solve the problem. These include, a heuristic for setting the targets at each layer, using a soft hinge loss, mini-batch training and such. Using these modifications the authors propose an algorithm (Algorithm 2 in appendix) to train such models efficiently. They compare the performance of a bunch of models trained by their algorithm against the ones trained using straight-through-estimator (SSTE) on a couple of datasets, namely, CIFAR-10 and ImageNet. They show superiority of their algorithm over SSTE.

I thought the paper is very well written and provides a really nice exposition of the problem of training deep networks with hard thresholds. The authors formulation of the problem as one of combinatorial optimization and proposing Algorithm 1 is also quite interesting. The results are moderately convincing in favor of the proposed approach. Though a disclaimer here is that I'm not 100% sure that SSTE is the state of the art for this problem. Overall i like the originality of the paper and feel that it has a potential of reasonable impact within the research community.

There are a few flaws/weaknesses in the paper though, making it somewhat lose.
- The authors start of by posing the problem as a clean combinatorial optimization problem and propose Algorithm 1. Realizing the limitations of the proposed algorithm, given the assumptions under which it was conceived in, the authors relax those assumptions in the couple of paragraphs before section 3.1 and pretty much throw away all the nice guarantees, such as checks for feasibility, discussed earlier.
- The result of this is another algorithm (I guess the main result of the paper), which is strangely presented in the appendix as opposed to the main text, which has no such guarantees.
- There is no theoretical proof that the heuristic for setting the target is a good one, other than a rough intuition
- The authors do not discuss at all the impact on generalization ability of the model trained using the proposed approach. The entire discussion revolves around fitting the training set and somehow magically everything seem to generalize and not overfit.

---

> ### Author Response · Authors · 2017-12-19
> **Response to Reviewer 3**
>
> Thank you for your review. We respond to each of your questions and comments below.
>
> Based on the quantization literature and other hard-threshold papers that we looked at and cited, SSTE is (by far) the most widely used method. It’s true that there are many variations of the straight-through estimator (STE), but we compare to the main one (SSTE), and don’t know of any that outperform SSTE. Note that neither STE nor SSTE has convergence guarantees (they’re biased estimators) but SSTE at least works well in practice.
>
> While we agree that it would be nice to have better guarantees for FTPROP-MB, it is typical in AI and combinatorial search (as you likely know) to start from a theoretically-justified approach and then use that to define a more heuristic approach that sacrifices those guarantees in favor of efficiently achieving the desired property (i.e., feasibility), as we do here. Since the problem we are solving is NP-complete and (most likely) hard to approximate, heuristics are unavoidable. By using the (soft) hinge loss at each layer, FTPROP-MB is implicitly trying to maximize “soft feasibility” of the network because of the correspondence between the hinge loss and margin maximization.
>
> Further, while feasibility is important for understanding the solution we propose, giving it up is necessary to avoid overfitting. This is similar to the linear-separability property of the perceptron where the robust method for learning a perceptron is to use a hinge loss instead of the perceptron criterion.  We intend to further study the properties of FTPROP and (soft) feasibility in the future.
>
> We did not put the FTPROP-MB pseudocode in the main paper because it’s such a simple algorithm and we were running short on space, but we can move it to the main body.
>
> Space limitations also precluded further discussions of generalization ability. We used standard approaches to avoid overfitting (L2 regularization, mini-batching, hinge vs. perceptron criterion, etc.), which we mention in the paper (but can make more clear) and which account for the good generalization performance.

---

### Decision · Program_Chairs · 2018-01-29
**ICLR 2018 Conference Acceptance Decision**

**Decision:**

Accept (Poster)

**Comment:**

The submission proposes optimization with hard-threshold activations.  This setting can lead to compressed networks, and is therefore an interesting setting if learning can be achieved feasibly.  This leads to a combinatorial optimization problem due to the non-differentiability of the non-linearity.  The submission proceeds to analyze the resulting problem and propose an algorithm for its optimization.

Results show slight improvement over a recent variant of straight-through estimation (Hinton 2012, Bengio et al. 2013), called saturated straight-through estimation (Hubara et al., 2016).  Although the improvements are somewhat modest, the submission is interesting for its framing of an important problem and improvement over a popular setting.